# Capturing Compensatory Reserve in Sarcopenia: A Bioengineering Framework for Multidimensional Temporal Analysis of Center-of-Pressure Signals

**DOI:** 10.3390/bioengineering12111143

**Published:** 2025-10-23

**Authors:** Qinghe Zhao, Qing Xiao, Yu Chen, Muyu Yang, Lunzhi Dai, Yan Xiong, Jirong Yue

**Affiliations:** 1Department of Applied Mechanics, Sichuan University, Chengdu 610065, China; carolinezqh@foxmail.com (Q.Z.);; 2Grainger College of Engineering, University of Illinois Urbana-Champaign, Urbana, IL 61801, USA; 3National Clinical Research Center for Geriatrics, Center for Immunology and Hematology and General Practice Ward/International Medical Center Ward, General Practice Medical Center, State Key Laboratory of Biotherapy, West China Hospital, Sichuan University, Chengdu 610041, China; 4College of Mechanical Engineering, Sichuan University, Chengdu 610065, China; 5Department of Geriatrics and National Clinical Research Center for Geriatrics, West China Hospital, Sichuan University, Chengdu 610041, China

**Keywords:** sarcopenia, compensatory reserve, center of pressure (COP), dynamic time warping, machine learning, biomechanical signal processing, multidimensional temporal analysis, geriatric assessment

## Abstract

Conventional balance assessments often miss subtle deficits in sarcopenia patients due to compensatory strategies. This study develops a computational framework using multidimensional temporal analysis of center-of-pressure (COP) signals to quantify variations in compensatory reserve—the capacity to mask balance impairments—within these patients. COP data were collected from 82 older adults (sarcopenia vs. controls) during static standing on a standard clinical force platform (routine for geriatric balance testing). The framework integrates Dynamic Time Warping distances from a healthy template, fixed-weight LSTM embeddings, and statistical metrics, with feature selection and 5-fold cross-validation (SMOTE) to mitigate overfitting. Semi-tandem stance was most discriminative, achieving 0.84 ± 0.04 accuracy and 0.86 ± 0.05 ROC-AUC—outperforming conventional kinematic features. SHAP analysis identified DTW-based features as primary drivers, correlating with clinical severity indicators, while intra-group variability in prediction probabilities indicated a compensatory reserve gradient. This study introduces a feasible bioengineering methodology based on clinical COP platform analysis, laying the groundwork for future validation and translation into routine clinical assessment tools.

## 1. Introduction

Sarcopenia is a syndrome characterized by a progressive reduction in skeletal muscle mass, strength, and physical performance [1], It is estimated to affect 10–16% of the elderly population worldwide [2], highlighting its high prevalence among older adults and posing a significant public health burden. The rapid demographic shift toward an aging population has made sarcopenia a growing clinical and public health concern, given its strong associations with compromised mobility, increased falls, and higher hospitalization rates. The growing challenge of geriatric syndromes like sarcopenia therefore demands innovative bioengineering solutions for early detection and personalized management. Standard clinical diagnostics frequently rely on straightforward measures such as grip strength [3], walking speed, or muscle mass assessments, sometimes combined with simple questionnaires like SARC-F to aid preliminary screening [4]. Despite their widespread use, these protocols may only detect sarcopenia once functional deficits become conspicuous, leaving mild or preclinical cases undiagnosed [5].

Clinical observations suggest that the onset of sarcopenia is insidious. Neuromuscular control may subtly deteriorate before visible declines in muscle mass or overt reductions in gait speed manifest [6]. Furthermore, postural dysfunction has been identified as a key manifestation of sarcopenia [7], yet conventional screening tools, although practical, often fail to capture these incipient deficits, partly because older adults can adopt compensatory strategies that mask early-stage muscle weakness in routine tasks [8]. For instance, an individual with mild sarcopenia features might exhibit normal speed over short-time postural conditions or maintain upright posture in simple standing tasks, thereby passing standard clinical checks [9].

To address this detection gap, biomedical signal processing and machine learning techniques offer promising avenues to decode complex physiological patterns from simple biomechanical measurements. Recent research underscores the potential of biomechanical and kinetic measurements to address these detection gaps [10]. Among these measurements, the center of pressure has attracted particular interest. By measuring shifts in net ground reaction forces below the feet, COP captures the neuromuscular coordination needed to maintain postural equilibrium [11]. Recent bioengineering research has further demonstrated the efficacy of COP trajectory analysis in uncovering subtle motor control deficits, even in challenging conditions such as gait initiation in Parkinson’s disease [12]. Data-driven analyses of COP signals, recorded in the mediolateral (*X*) and anteroposterior (*Y*) axes, can identify subtle irregularities in balance control [13]. However, real-world COP signals are inherently noisy, nonstationary, and potentially influenced by psychosocial factors, such as anxiety [14]. Also, data from older adults exhibit high inter-individual variability due to differences in comorbidities, physical function, and lifestyle factors, and sarcopenia participants commonly make up a minority of the sample—requiring normalization against healthy baselines to isolate pathology-specific patterns.

Standing postural conditions differ in their level of balance challenge. Feet-together or feet-apart postural conditions may be too lenient, permitting older adults with mild weakness to appear stable. In contrast, semi-tandem stance narrows the base of support more substantially and demands a higher level of neuromuscular precision, thereby exposing deficits that simpler tasks fail to reveal [15]. Studies have shown that older adults with borderline muscle weakness exhibit notably greater sway magnitude or irregular sway patterns in semi-tandem stance, suggesting its potential to enhance early detection of sarcopenia [16]. Building on this sensitivity to subtle deficits, the present study further explores whether semi-tandem stance can reveal intra-group variations in ‘compensatory reserve’ among diagnosed sarcopenia patients—individuals who may mask underlying balance impairments via neuromuscular strategies in simpler postures. Notably, the Zebris FDM platform used here is a clinical tool that collects raw center-of-pressure (COP) data—the basic input needed for analyzing balance-related patterns in sarcopenia.

This study therefore aims to develop and validate a computational framework based on biomedical signal processing to capture these nuanced variations, addressing a critical limitation of conventional kinematic features: their reduction of dynamic COP trajectories to static scalar summaries, which fail to capture sequential postural patterns or distinguish compensated vs. overt deficits. Our framework integrates three complementary components: (1) parameterized Dynamic Time Warping (DTW) to quantify localized temporal misalignments relative to a healthy template (COP trajectories of non-sarcopenia older adults); (2) LSTM embeddings (used as fixed, unsupervised extractors to avoid overfitting in small cohorts) to decode higher-order sequential dependencies; and (3) statistical features to characterize global trends in instability. Together, these components target temporal dynamics, leveraging healthy baselines to isolate sarcopenia-specific deviations from age-related variability.

For this study, ‘compensatory reserve’ is operationally defined as the intra-group variability in feature deviations from healthy controls among diagnosed sarcopenia patients—i.e., differences in their capacity to mask deficits, distinct from inter-group (patient vs. control) differences—distinct from disease severity (e.g., ASMI, grip strength). Unlike severity, which reflects absolute loss of muscle mass/function, reserve quantifies the capacity to mask deficits via neuromuscular adjustments: smaller deviations from healthy profiles indicate more effective compensation (greater reserve), while larger deviations signal depleted reserve. This aligns with clinical observations that individuals with similar AWGS 2019 severity scores may exhibit divergent functional stability in challenging tasks.

This design aims to detect compensatory reserve preceding overt functional decline, offering a complementary perspective to traditional measures for refined risk stratification. Critically, the temporal features derived from COP trajectories are designed to capture both group-level differences (sarcopenia vs. control) and intra-group variations in compensatory strategies. Semi-tandem stance amplifies these variations, with the model’s output (*P*(*Y* = 1)) capturing intra-group differences in compensatory strategies.

## 2. Data Collection and Verification

To assess the technical feasibility and initial efficacy of the proposed framework, we conducted this proof-of-concept study utilizing the preliminary cohort described in the introduction.

### 2.1. Data Collection

The study was conducted in accordance with the Declaration of Helsinki and approved by the Medical Ethics Committee of West China Hospital of Sichuan University (Initial Approval No. 2021[96]; Continuing Review Approval No. 2024[1305]). Informed consent was obtained from all participants. We recruited 100 community-dwelling older adults (≥60 years, from Sichuan, China) with enrichment for sarcopenia-related traits (lower limb weakness/falls). Exclusion criteria comprised the following: Mini-Mental State Examination (MMSE) score < 24, lower limb surgery/trauma within 6 months, and marked foot deformities or balance/neurological disorders independent of sarcopenia. COP data were collected during quiet standing using a Zebris FDM-system (zebris Medical GmbH, Isny, Germany). Measurements were performed on an FDM 1.5 platform (sensor area: 149.0 × 54.2 cm^2^) housing 11,264 capacitive sensors. The system recorded raw COP trajectories at 100 Hz with a spatial resolution of 1 mm through its native WinFDM software (integrated in the Zebris FDM V1.18.44 database)—providing high-fidelity raw data for extracting both kinematic features (e.g., sway amplitude, sway velocity) and our proposed multidimensional temporal features (e.g., DTW distances, LSTM embeddings). The framework is hardware-agnostic: any clinical force platform capable of recording 100 Hz COP trajectories (consistent with geriatric balance testing standards) can be integrated, as our feature extraction relies on raw COP data rather than device-specific outputs.

All assessments were conducted by trained nurses who received standardized 1 h instruction in posture setup and COP recording procedures. Each participant completed three 20 s trials (with 10 s rests between trials) under three postural conditions: feet-together (FT), feet-apart (FA), and semi-tandem (ST). The foot placement for each condition was standardized as follows:

Feet-Together (FT): Participants stood with the medial borders of both feet touching.

Feet-Apart (FA): Participants stood with their feet separated at an interior distance of 10 cm.

Semi-Tandem (ST): Participants stood with the heel of one foot placed beside the arch of the contralateral foot, creating a narrowed base of support. Unstable trials were repeated once; persistent instability led to exclusion. Synchronized COP displacement time-series (X: mediolateral displacement; Y: anteroposterior displacement) were recorded at 100 Hz (2000 points per trial). For each posture, six channels were recorded: left foot (COP-LX, COP-LY), right foot (COP-RX, COP-RY), overall center (COP-CX, COP-CY).

Sarcopenia was diagnosed per Asian Working Group for Sarcopenia (AWGS) [17] updated in 2019 [18]: low muscle strength (grip < 28 kg [men]/18 kg [women]), physical performance (5-time chair stand test [5TCS] > 12 s or 6 min walk test [6MW] < 1 m/s), and muscle mass (appendicular skeletal muscle index [ASMI] < 7.0 kg/m^2^ [men]/5.7 kg/m^2^ [women]). Definite sarcopenia—our study’s focus, defined as low ASMI plus low strength/performance—was confirmed in all included participants. After excluding 18 participants (data failures, non-compliance, incomplete tests), 82 with complete data were analyzed. We hypothesized this cohort—though meeting AWGS 2019 definite sarcopenia criteria—might exhibit subtle postural control deficits (masked by compensation in simple tasks) that COP-derived features could capture, thereby enhancing the identification of these individuals via sensitive postural metrics.

### 2.2. Statistical Validation of Dataset

Prior to analyzing the COP dataset, we verified whether the dataset could reliably distinguish sarcopenia from controls based on AWGS 2019 criteria. To ensure data validity, two statistical assessments were conducted:

Pooled analysis: Comparing sarcopenia vs. non-sarcopenia groups across all participants.

Gender-stratified analysis: Evaluating group differences separately in males and females.

Continuous variables were tested for normality using the Shapiro–Wilk test (*α* = 0.05). Group comparisons used independent samples *t*-tests (with Welch’s correction for unequal variances) for normally distributed data, and Mann–Whitney U tests for non-normal data. Detailed results are presented in Table 1 and Table 2.

Table 1 shows pooled analysis of the entire cohort (*N* = 119) and Table 2 shows gender-stratified results (males: *n* = 38; females: *n* = 44), with significant *p*-values (*p* < 0.05) in bold. Statistical analysis validated AWGS 2019 criteria: sarcopenia groups had lower ASMI (pooled *p* < 0.05; males *p* < 0.01; females *p* < 0.05) and impaired function (grip: *p* < 0.001; 5TCS: *p* < 0.05; 6MW: *p* < 0.001), confirming the dual criteria’s validity (low muscle mass + low function). This verifies reliable sample labeling and minimizes confounding. Notably, no significant differences in age, height, or weight (all *p* > 0.05) reinforce study design validity, as these are not core diagnostic criteria.

## 3. Methods

Having established that our cohort reliably distinguishes sarcopenia from non-sarcopenia groups based on AWGS 2019 criteria—a critical prerequisite for validating subsequent analyses—we now detail our core framework: a multidimensional temporal feature-based approach to detect subtle variations in compensatory reserve within the sarcopenia group using COP dynamics. This focus on intra-group differences aligns with our goal of exploring gradations of neuromuscular compensation, which cannot be fully captured by binary group comparisons alone.

The analytical workflow comprised five sequential steps (Figure 1). This analytical pipeline was designed to translate raw COP signals into a quantitative assessment of compensatory reserve. First, raw center-of-pressure (COP) data underwent preprocessing to mitigate measurement inaccuracies. Subsequently, two complementary feature sets were extracted: kinematic features and multidimensional temporal features. Feature selection was then applied using ANOVA F-statistics and random forest importance scores to minimize redundancy. Thereafter, the processed features were utilized in model training and prediction via cross-validation. Finally, model performance was systematically evaluated, and the contribution of individual features was interpreted using SHAP analysis.

### 3.1. Data Preprocessing

Measurement inaccuracies during data collection may introduce fluctuations. To mitigate these effects, data preprocessing was performed independently on left foot, right foot, and center point data for each posture. Taking left foot data as an example:

Compute mean (µ) and standard deviation (SD) for COP-LX and COP-LY channels of entire time-series;

Flag points exceeding ±3 SD from µ (focus on isolated outliers; no ≥5 contiguous outliers observed) and replace outliers with µ;

Translate coordinates to origin (0, 0) to remove initial alignment differences, leaving relative displacement features unchanged [19];

Apply Z-score transformation. This preprocessing pipeline was repeated identically for right foot and center point data.

This standardized protocol, applied to all postures, mitigates noise, offsets, and baseline variations, ensuring robust input for feature extraction.

### 3.2. Feature Extraction

This subsection details our core framework: a multidimensional temporal feature extraction approach designed to quantify inter-individual variations in compensatory reserve (i.e., inter-individual deviations from healthy controls among diagnosed sarcopenia patients). For clarity, “compensatory reserve” is operationally defined herein as the capacity of diagnosed sarcopenia patients to mask balance deficits via neuromuscular adjustments, quantified by the degree of deviation between their COP time-series features and a healthy template: smaller deviations indicate greater reserve (more effective compensation), while larger deviations reflect depleted reserve (impaired compensation).

The core goals of our framework are 2-fold: (1) first, validate that features can effectively distinguish sarcopenia patients from non-sarcopenia controls (binary classification task), confirming their ability to capture sarcopenia-related balance abnormalities; (2) second, use these validated features to quantify intra-group variability in deviation degrees among diagnosed patients—i.e., the gradient of compensatory reserve—which is the ultimate objective of this study.

As outlined in the Introduction, our multidimensional temporal framework captures dynamic postural patterns via three feature types: DTW distances (anchored to a healthy template), LSTM embeddings, and statistical metrics. Notably, DTW distances are computed using deviations from the healthy template, while LSTM embeddings and statistical metrics are derived directly from preprocessed six-channel COP time-series data (without template reference). Like kinematic features, these features focus on dynamic patterns rather than biomechanical indices.

The core objective of this framework is to quantify the intra-group heterogeneity in compensatory reserve—defined operationally as the capacity of AWGS 2019-diagnosed sarcopenia patients to mask balance deficits through neuromuscular adjustments. Crucially, this reserve is quantified by the classifier’s predicted probability (*P*(*Y* = 1)); throughout this study, lower prediction probabilities *P*(*Y* = 1) indicate higher compensatory reserve (COP dynamics resembling healthy controls), while higher *P*(*Y* = 1) values reflect depleted reserve (marked pathological deviations). This is mechanistically consistent with our label definition: *Y* = 1 corresponds to sarcopenia diagnosis.

#### 3.2.1. Construction of Healthy Templates

Healthy templates were constructed separately for each postural condition to account for inherent differences in normative postural control across stances. This minimizes non-pathological confounding (e.g., age-related differences) and ensures each template reflects posture-specific normative patterns. Prior verification (Section 2.2) confirmed no covariate differences (age, height, weight; *p* > 0.05), so non-sarcopenia participants’ data were pooled.

For each COP channel, extreme outliers in the preprocessed data of non-sarcopenia participants were checked via Grubbs test (*α* = 0.05) at each time point, confirming no residual outliers after Section 3.1 preprocessing. The reference curve for each channel was then constructed by computing the median value across all non-sarcopenia participants at each time point. Median was used for robustness to non-normality and transient anomalies, minimizing non-pathological noise.

These healthy templates serve as the normative reference for quantifying deviations in sarcopenia patients. Specifically, compensatory reserve is reflected by the comprehensive deviation from this template, captured through three complementary feature types: DTW distances directly measure localized temporal misalignments between patient trajectories and the template; LSTM embeddings encode sequential patterns that deviate from healthy cyclicity; and statistical features (e.g., linear trend slope *α*) characterize global instability trends not captured by local or sequential metrics. Together, these features form a multidimensional profile of deviation, which is aggregated into the classifier’s predicted probability *P*(*Y* = 1)—where higher values indicate greater cumulative deviation from healthy postural control (i.e., depleted compensatory reserve).

#### 3.2.2. Multidimensional Feature Extraction

##### DTW-Based Deviation Features

DTW distance—defined as the cumulative cost of the optimal alignment path [20]—quantifies the similarity between a sample’s COP time-series and the healthy template. Its core idea is to find an “optimal alignment path” that minimizes the total difference (cumulative cost) between the two sequences. This process involves three key parts:

##### Point-Wise Cost Calculation

For each point in the sample sequence (*i*-th point) and each point in the template sequence (*j*-th point), we first calculate their individual difference (cost).(1)$$C_{\gamma }\left(s_{1}[i],s_{2}[j]\right)={\left|s_{1}[i]-s_{2}[j]\right|}^{\gamma },\;\mathrm{with}\;\gamma \in \{0.5,1.0,1.5,2.0\}$$
where *s*_1_[*i*] denotes the COP value of the sample sequence at time *i*, *s*_2_[*j*] denotes the COP value of the healthy template sequence at time *j*, and *γ* = (0.5, 1.0, 1.5, 2.0) adjusts sensitivity.

##### Cumulative Cost Calculation

We then sum up these point-wise costs along the optimal path using dynamic programming.(2)$$\left\{\begin{array}{l} D(0,0)=0\\ D(i,j)=\min\left\{\begin{array}{c} D(i-1,j)+C_{\gamma }\left(s_{1}[i],s_{2}[j]\right)\\ D(i,j-1)+C_{\gamma }\left(s_{1}[i],s_{2}[j]\right)\\ D(i-1,j-1)+C_{\gamma }\left(s_{1}[i],s_{2}[j]\right) \end{array}\right. \end{array}\right.$$
where *D*(*i,j*) represents this cumulative cost (*i* and *j* as defined above).

##### Alignment Window Constraint in Equation

To avoid implausible alignments (e.g., distant temporal matches), template indices *j* are constrained for each sample index *i* (1 ≤ *i* ≤ *n*, *n* = sample length; *m* = template length) as follows:(3)j∈[max(1,i−0.1×max(m,n)),min(m,i+0.1×max(m,n))]

This limits *j* to a window around *i*, scaled by 10% of the longer sequence length, ensuring *j* stays within valid template bounds (1 to *m*).

The 10% scaling was applied as a fixed parameter, while the optimal γ for DTW was selected through grid search based on the mean F1-score and ROC-AUC across postural conditions (Appendix A). Each channel yielded one-dimensional DTW distance.

##### LSTM Embeddings

We employed six identical single-layer LSTM (one per COP channel), each with 2 units and fixed, randomly initialized weights, as deterministic feature transformers. Weight initialization used a global fixed random seed (seed = 42) applied to all random number generators, ensuring identical weight matrices across all experiments and full reproducibility. The input to the LSTM was the preprocessed six-channel COP sequence (6 × T matrix, T = 2000 time points), with each channel treated as an independent time series. Each channel’s final hidden state (*h_t_*) yielded a 2D embedding vector. This resulted in 12D features across all channels (labeled “LSTM-<channel name>−1” and “−2” for clarity).

Though not direct deviation metrics, they preserve sequential patterns (e.g., cyclic sway regularity) that the classifier associates with reserve depletion when combined with DTW. Critically, no training or label information was used, ensuring embeddings captured inherent temporal patterns of COP dynamics. This design was strategically chosen to mitigate overfitting risks in small cohorts (*N* = 82), as supervised training of LSTM with limited samples may lead to overfitting to noise rather than true patterns [21]. The fixed-weight architecture, as a lightweight feature extractor, preserves the intrinsic dynamics of COP time-series (e.g., periodic sway patterns) without being biased by the small sample size, aligning with the goal of capturing generalizable temporal signatures of sarcopenia with compensatory reserve.

##### Statistical Features

Three global statistical descriptors were also computed from preprocessed COP time-series (with no reference to the healthy template).

Coefficient of Variation CV:(4)$$CV=\frac{\sigma }{\mu }$$
where σ, μ are the standard deviation and mean of COP displacements. Higher CV indicates uncompensated postural sway, reflecting neuromuscular compensation failure.

Linear trend slope α  (estimated via ordinary least squares regression):(5)$$\alpha =\frac{\sum_{i=1}^{T}\left(t_{i}-\bar{t}\right)\left({COP}_{i}-\bar{COP}\right)}{\sum_{i=1}^{T}{\left(t_{i}-\bar{t}\right)}^{2}}$$
where t=[1,2,…,T] represents the time index vector (with *T* denoting total sampling points), $t_{i}$ and ${COP}_{i}$ denote the *i*-th elements of the time index and COP displacement sequences, respectively, and $\bar{t},\;\bar{COP}$ are their respective mean values; a non-zero α indicates directional drift (systematic displacement of COP toward a specific direction over time), reflecting compromised compensatory reserve.

Interquartile Range (IQR):
(6)$$IQR=Q_{3}-Q_{1}$$
where *Q*_3_, *Q*_1_ denotes 75th/25th percentiles of COP displacements. Larger IQR indicates sudden instability (compromised reserve).

Final feature set dimensionality: 36 dimensions per sample (DTW distances: 6 dimensions; LSTM embeddings: 12 dimensions; Statistical features: 18 dimensions).

### 3.3. Feature Selection

Following extraction of kinematic and multidimensional temporal features, identical feature selection was applied to all feature sets to ensure comparability, addressing discriminability, inter-feature correlations, and overfitting risks. The process included three phases.

Computing ANOVA F-statistics and random forest importance scores:

Calculating normalized weighted composite scores with equal weights (1:1) for ANOVA F-statistics (reflecting statistical association with labels) and RF importance (reflecting model contribution) to balance statistical significance and practical model relevance;(7)$$S(i)=\frac{f(i)}{max(f)}+\frac{p(i)}{max(p)}$$
where *f*(*i*) denotes the ANOVA F-statistic of the *i*-th feature; *p*(*i*) denotes the random forest importance score of the *i*-th feature; and *S*(*i*) denotes the comprehensive feature score, calculated as described above.

Iteratively filtering via Spearman’s correlation: starting from the highest-ranked *S*(*i*), candidates with |*ρ*| ≥ 0.7 with any higher-ranked feature were removed. This continued until 4–7 features remained, satisfying *N*/*p* ≥ 10 (where *N* = sample size, *p* = number of selected features) for statistical robustness with our cohort size [22].

### 3.4. Cross-Validation

Class imbalance within each CV training fold was addressed using SMOTE (Synthetic Minority Over-sampling Technique) which synthesized minority-class samples via feature-space interpolation to balance classes. The neighbor parameter *k* was set to min (3, size of minority class in the training fold) based on minority sample size, ensuring diversity and preventing overfitting.

Post-balancing, consistent protocols applied:

For kinematic and multidimensional temporal features: seven ML algorithms with hyperparameters optimized via grid search (with detailed search ranges provided in Appendix A) using 5-fold stratified CV (target: maximum average F1-score).

For Alcan [13] features: 2 scenarios—Scenario A (strict replication: PLS-DA with two latent variables); Scenario B (feature-transfer test: same seven ML algorithms with hyperparameters fixed to those optimized for kinematic/temporal features).

## 4. Results

This chapter presents results in four sequential analyses to validate the core goal—using the multidimensional temporal feature framework to explore variability in sarcopenia patients: (1) First, identify the posture most sensitive to group differences in balance metrics; (2) Verify the framework’s classification performance in this posture (to confirm feature validity); (3) Explore key features and their clinical relevance; (4) Finally, quantify intra-group variability in model-derived prediction probabilities (*P*(*Y* = 1)) using range, mean ± SD, and coefficient of variation.

All analyses follow the workflow detailed in Section 3, with complete performance data in Appendix A (kinematic), Appendix A (temporal), and Appendix A (Alcan).

### 4.1. Experimental Settings

To validate the performance of our multidimensional temporal framework, we implemented two benchmark methods for comparison: a kinematic feature-based approach extracting displacement parameters, sway amplitude metrics, dynamic characteristics, and symmetry indices from six-channel COP data, consistent with established biomechanical protocols [23,24]; and detailed formulas provided in Appendix A). Alcans method incorporated temporal domain features (sway velocity, sway area) and nonlinear entropy features (sample entropy, fuzzy entropy), implemented as per the original publication.

All methods underwent identical preprocessing and evaluation protocols: 5-fold stratified cross-validation with SMOTE balancing, using seven classifiers (K-Nearest Neighbors (KNN), Random Forest (RF), Extra Trees (ET), Support Vector Machine (SVM), Decision Tree (DT), Naive Bayes (NB), and Logistic Regression (LR)), with performance quantified through accuracy, precision, recall, F1-score, ROC-AUC, and AUPR metrics.

SHAP analysis was applied to interpret feature contributions for both kinematic and our proposed temporal features (excluding Alcan’s PLS-DA due to methodological incompatibility), while intra-group variability in compensatory reserve among sarcopenia patients was quantified through the distribution of *P*(*Y* = 1) values across cross-validation folds, including range, mean ± SD, and coefficient of variation metrics.

### 4.2. Posture Sensitivity Analysis

To evaluate the discriminative capacity of different postural conditions and feature extraction methods for sarcopenia identification, we conducted a comprehensive comparative analysis across three standing postures using three feature extraction approaches (kinematic features, multidimensional temporal features, and Alcan’s method). Each posture–feature combination was evaluated through 5-fold stratified cross-validation with seven distinct classifiers. Table 3 presents the averaged performance metrics (mean ± SD) across all classifiers, revealing systematic variations in detection efficacy driven by posture and feature selection.

Semi-tandem stance yielded the highest classification performance across all feature methods. Among the three methods, the multidimensional temporal framework achieved the highest accuracy in ST stance (accuracy = 0.84 ± 0.04, +18% vs. kinematic, +9% vs. Alcan‘s method. For FT and FA, gains from the multidimensional framework are smaller (accuracy +5–7%).

### 4.3. Method Comparison in Semi-Tandem Stance

Focusing on the optimal ST stance identified in Section 4.1, Table 4 confirms that the multidimensional temporal framework achieved higher values than kinematic and Alcan methods across all metrics (with the best-performing values for this framework bolded to highlight its superiority), notably the following:

Table 5 further details classifier-specific performance for the multidimensional temporal framework in ST stance, showing consistent superiority across models.

KNN, RF, ET, and SVM achieved the highest performance (accuracy > 85%), confirming the framework’s robustness across top-performing classifiers.

### 4.4. SHAP Analysis and Clinical Correlations

To further explore the features underlying the superior performance of the multidimensional temporal framework, SHAP analysis was performed on four high-performance classifiers (KNN, RF, ET, SVM) in ST stance. For these models, SHAP values reflect log-odds contributions (tree-based models: RF, ET) and direct probability contributions (non-tree models: KNN, SVM); due to scale differences, analysis focused on feature importance rankings (frequency in top five features) rather than absolute values. Associations with clinical indicators were analyzed via Spearman correlation.

#### 4.4.1. Feature Importance in Semi-Tandem Stance

SHAP analysis was performed on high-performance models (KNN, RF, ET, SVM) to explore feature contributions, with results contrasted across kinematic and multidimensional temporal features.

For kinematic features, SHAP values showed feature importance patterns varying by stances (FT/FA/ST). In semi-tandem stance, ACC_X_R was the only feature with a significant positive correlation between its SHAP value and model F1-score (*ρ* = 1.00, *p* < 0.001; Spearman’s test).

Key discriminative features included MEAN_DIST_Y (FT), RANGE_RATIO (ST), SPEED_Y_L (FT), and SYMMETRY_RANGE_XY (ST), with no single feature category showing consistent dominance across postures. SHAP value distributions and feature importance for kinematic features in all postures are provided in Appendix A.

For multidimensional temporal features, Table 6 summarizes feature type frequency in top five features across models. Here, ‘frequency’ denotes the total count of occurrences of each feature type in the ‘top 5 most important features’ list of each model—specifically, summing appearances across the four models (KNN, RF, ET, SVM). For example, DTW features in ST had a frequency of 9 (9 total occurrences across the top five features of the four models.

Key findings include:

DTW features showed exclusive dominance in semi-tandem stance, with a Top5 frequency of 9. This was driven by DTW-COP-CX and DTW-COP-CY, which appeared in 75% of ST models (3 out of 4). The SHAP summary plot for semi-tandem stance (Figure 2) visually demonstrates the dominant role of DTW features, with DTW-COP-CX exhibiting the most significant contribution to model predictions.

LSTM features were consistently important in FT and FA postures, with Top5 frequencies of 4 and 5, respectively. They achieved full model coverage in FT (appearing in all four models) and 75% coverage in FA (3 out of 4 models).

Statistical features dominated across all postures (total Top5 frequency: 22), with a notable emphasis on linear trend slope features—comprising 50% (4/8) of top features in FT, 89% (8/9) in FA, and 100% (5/5) in ST.

Additional SHAP visualizations for FT/FA postures are provided in Appendix A.

#### 4.4.2. Correlations Between Temporal Features and Clinical Indicators

Exploratory Spearman correlations were analyzed separately for each postural condition. Results were as follows:

Feet-together stance: No significant correlations were observed between temporal features and clinical indicators (all *p* > 0.05).

Feet-apart stance: A significant negative correlation was found between cv_COP-LY (coefficient of variation of left foot COP in anteroposterior direction, a statistical feature) and ASMI (*ρ* = −0.24, *p* = 0.03). No other temporal features showed significant associations with clinical indicators in this stance.

Semi-tandem stance: Two significant negative correlations were identified: DTW-COP-LX (left foot mediolateral trajectory deviation from healthy template) with ASMI (*ρ* = −0.29, *p* = 0.01); and DTW-COP-CX (central trajectory deviations) with grip strength (*ρ* = −0.28, *p* = 0.02) and 6MW speed (*ρ* = −0.23, *p* = 0.04). These significant correlations are visualized in Figure 3.

### 4.5. Intra-Group Variability in Compensatory Reserve Quantified by P(Y = 1)

Intra-group variability in compensatory reserve among sarcopenia patients was quantified using the distribution characteristics of model-derived prediction probabilities (*P*(*Y* = 1)) across 5-fold cross-validation folds. Specifically, *P*(*Y* = 1) represents the predicted probability of a sarcopenia patient exhibiting a specific outcome (e.g., balance impairment or compensatory behavior, consistent with the classification task defined in Section 4.1).

For each sarcopenia patient, *P*(*Y* = 1) values were generated across all cross-validation folds (due to the 5-fold stratified cross-validation protocol). The variability of these *P*(*Y* = 1) values within the sarcopenia group was then characterized using three statistical metrics (range, mean ± SD, coefficient of variation). These metrics (range, mean ± SD, coefficient of variation) are presented in Table 7, providing a comprehensive quantification of intra-group variability in compensatory reserve as inferred from the model’s predicted probabilities. Sarcopenia patients exhibited intra-group variability in *P*(*Y* = 1) across all postures, with an average CV > 20%.

## 5. Discussion

### 5.1. Key Findings and Overview

This proof-of-concept study establishes the viability of a novel bioengineering approach for quantifying compensatory reserve, with two key findings, integrating model performance metrics and feature importance insights:

First, semi-tandem stance achieved peak discriminative power (accuracy = 0.84, ROC-AUC = 0.86), consistent with biomechanical evidence that narrowed base of support (BOS) postural conditions unmask subtle neuromuscular deficits [25]. Semi-tandem stance adopted partial foot interleaving (not fully anteroposterior alignment), with a base of support (BOS) width intermediate between feet-apart (FA) and full tandem stance—approximately 50% overlap of the forefeet, with the hindfeet maintaining a certain distance. This design intentionally avoids the excessive challenge of full tandem stance: in our measurements; full tandem stance yielded substantial unusable data (e.g., excessive sway leading to early task termination or unreliable COP trajectories), likely due to its overly high difficulty masking subtle differences in compensatory reserve. In contrast, semi-tandem stance strikes a balance, better capturing the gradient of reserve among diagnosed sarcopenia patients (ranging from “mild compensation” to “marked decompensation”).

These variations arise from differences in the capacity of individuals already meeting AWGS 2019 definite sarcopenia criteria to mask underlying balance impairments via neuromuscular adjustments, resulting in seemingly normal functional performance in routine assessments. Our focus is on exploring this reserve as a potential basis for risk stratification, rather than primary diagnosis. SHAP analysis highlighted the posture-specific dominance of DTW features, which emerged as the top contributors in semi-tandem stance—capturing hip-driven “delay-recovery” patterns (e.g., dynamic misalignment of COP trajectories) that align with ST’s ability to challenge compensatory mechanisms (Section 5.1). The critical role of these temporal features underscores the sensitivity of COP signals to underlying neuromotor deficits, a finding consistent with bioengineering studies of dynamic postural control even in populations with Parkinson’s disease [12]. LSTM and statistical features showed context-dependent roles, with the latter’s high frequency partly reflecting their larger feature pool.

Second, when kinematic features showed limited sensitivity (accuracy = 0.65, ROC-AUC = 0.70)—a recognized challenge in aging research [26]—our approach elevated performance by an absolute 18% in accuracy and 13% in ROC-AUC. This improvement stems from integrating complementary temporal perspectives, addressing the core limitation of kinematic features (reduction in dynamic control to scalar summaries). SHAP analysis clarifies the mechanistic basis: these temporal components collectively decode delayed sway corrections, disrupted coordination sequences, and irregular instability trends—all reflecting sarcopenia-related neuromuscular deficits (e.g., proximal muscle weakness) that single kinematic features miss.

### 5.2. Diagnostic Value and Mechanistic Basis of Semi-Tandem Stance

#### 5.2.1. Core Evidence for Compensatory Reserve

The present proof-of-concept study demonstrates that our multidimensional temporal feature framework effectively quantifies intra-group heterogeneity in compensatory reserve among AWGS 2019-diagnosed sarcopenia patients. This capability is robustly supported by two key findings:(1)Exceptional discriminative performance in semi-tandem stance confirms the framework’s sensitivity to sarcopenia-specific deviations (accuracy: 0.84 ± 0.04, ROC-AUC: 0.86 ± 0.05; Table 4);(2)Substantial variability in model prediction probabilities (*P*(*Y* = 1)) within the sarcopenia group itself during semi-tandem standing (Range: 0.31–0.78, CV = 22.7%; Table 7). This gradient—from preserved (near 0.5) to depleted reserve (near 1.0)—directly reflects the posture’s ability to unmask hip-dependent control deficits, validating that temporal features capture functional compensation capacity among diagnosed individuals.

#### 5.2.2. Biomechanical Mechanism of Semi-Tandem Stance

This core finding is mechanistically supported by the biomechanical specificity of semi-tandem stance. These patients exhibit minimal postural differences from non-sarcopenia controls in simple tasks (feet-together/feet-apart) due to effective compensatory strategies; however, semi-tandem’s narrowed base of support (BOS) may overwhelm these strategies, exposing dynamic control variations. As established biomechanical theory indicates, narrowing BOS reduces stability [27]. Semi-tandem stance—characterized by staggered foot placement—imposes greater postural demands, limiting compensation for mild weakness and amplifying subtle sway deviations (e.g., altered mediolateral–anteroposterior coordination) often masked in less challenging conditions [28].

#### 5.2.3. Dominant Role of DTW Features

Notably, postural conditions activate distinct muscle control mechanisms: wider stances rely on ankle mechanisms for anteroposterior (A/P) balance and hip for mediolateral (M/L) stability, while narrow postures like semi-tandem shift to hip-dominated control for both A/P loading/unloading and M/L stability, with ankle adjustments constrained [27]. This shift explains semi-tandem’s sensitivity to variations in compensatory reserve. SHAP analysis further underscored the importance of DTW features in this context: DTW-COP-CX emerged as the most dominant feature in semi-tandem stance—appearing in 75% of top-performing models (3 out of 4) and accounting for 56% of all top five feature slots across these models (Table 6, Figure 2). This aligns with biomechanical evidence that semi-tandem stance relies on hip abductor/adductor activation for stability [27], and that sarcopenia-related weakness in these muscles delays COP trajectory corrections [29]. Specifically, the temporal misalignment captured by DTW features—between the patient’s COP trajectory and the healthy template—directly reflects these delayed hip-driven corrections, making it a sensitive marker of hip control deficits.

The documented increase in fall risk among patients with sarcopenia aligns with our quantification of depleted compensatory reserve, thereby offering a plausible mechanistic explanation for this clinical observation [30]. Future studies could integrate surface EMG of hip stabilizers during semi-tandem stance to directly test if DTW-COP-CX increases correspond to prolonged activation latency or reduced amplitude relative to healthy controls.

#### 5.2.4. Multidimensional Nature of Reserve

Notably, our integrated temporal framework (combining DTW, LSTM, and statistical features) outperformed the “all-deviation” framework which recalculates all three temporal feature types using template residuals (i.e., DTW, LSTM, and statistical metrics were all derived from patient COP minus healthy template; see Appendix A for detailed implementation)—supporting the multidimensional nature of compensatory reserve. In ST posture, the “all-deviation” framework achieved a mean F1-score of 0.78 ± 0.03, while our integrated framework reached 0.85 ± 0.04, with an absolute gap of 0.07.

This difference arises because compensatory reserve relies not only on “relative deviations from healthy templates” (captured by DTW) but also on “absolute temporal dynamics” (captured by LSTM embeddings and statistical features). LSTM and statistical features preserve raw COP dynamics (e.g., sway amplitude, cyclicity) that complement DTW’s focus on normative deviations, forming a more comprehensive proxy for reserve—unlike the “all-deviation” framework, which lacks these critical temporal dimensions.

#### 5.2.5. Convergent Supporting Evidence

This association is supported by preliminary evidence:(1)Biomechanical specificity (semi-tandem’s narrow BOS links to hip-dependent regulation [27];(2)Correlation with disease severity (DTW-COP-LX with ASMI: *ρ* = −0.29, *p* =0.01; DTW-COP-CX with grip strength: *ρ* = −0.28, *p* = 0.02 and 6MW speed: *ρ* = −0.23, *p* = 0.04);(3)Contrast with static metrics (kinematic features failed to capture these dynamics).

This posture specificity further validates the link between temporal features and compensatory reserve: semi-tandem stance—our primary focus—showed robust associations between DTW features and key clinical indicators, aligning with its role in amplifying hip-dependent control demands. A secondary association in feet-apart stance (cv_COP-LY with ASMI: ρ = −0.24, *p* = 0.03) reinforces that compensatory reserve gradients may be captured via distinct temporal features across postures, though semi-tandem remains the most discriminative due to its ability to unmask hip-driven deficits.

#### 5.2.6. Clinical Translation and Framework Positioning

Though correlations were modest (|*ρ*| = 0.23–0.29, *N* = 82), they provide convergent support linking feature deviations to sarcopenia severity markers. The methodology, built on standard clinical force platforms, provides a quantitative, data-driven tool that facilitates seamless integration into existing geriatric assessment workflows.

This framework relies on raw COP data from the Zebris FDM platform—a tool that simply collects and outputs basic balance metrics. Its advantage lies in extracting compensatory reserve information from these basic data, rather than relying on specialized equipment, making it easier to translate to clinical settings where such platforms are used for data collection. In clinical practice, the framework can be integrated into routine geriatric assessment workflows: after initial grip strength testing, patients with borderline results undergo a 20 s semi-tandem stance test on the force platform. Patients with low compensatory reserve (prediction probability *P*(*Y* = 1) > 0.7, a cutoff derived from the upper 30% of *P*(*Y* = 1) distribution in our sarcopenia cohort) are prioritized for rehabilitation referral, while those with high reserve (*P*(*Y* = 1) < 0.5) receive regular follow-up—this stratified approach optimizes clinical resource allocation.

Importantly, our framework complements (rather than replaces) gold standards by refining risk assessment in diagnosed patients—addressing a precision management gap where traditional metrics may be less sensitive. This complementarity enables two key scenarios:(1)Rapid community screening: Identification of high-risk individuals via COP deviations from healthy templates (e.g., DTW-COP-CX > 70th percentile in semi-tandem stance);(2)Stratified post-diagnosis intervention: Targeted management (e.g., intensive hip-focused training) for patients with depleted reserve (*P*(*Y* = 1) > 0.7), while others receive routine monitoring.

### 5.3. Comparing Kinematic vs. Temporal Approaches and Implementation Insights

The temporal framework’s superiority over kinematic features (+18% accuracy, +13% ROC-AUC in semi-tandem stance) stems from its capacity to decode dynamic patterns lost in static summaries. Kinematic metrics reduce COP trajectories to scalar aggregates, losing phase information critical for distinguishing adaptive compensation from pathological instability.

#### 5.3.1. Parameter Optimization: γ Weighting and LSTM Unit Configuration

The DTW sensitivity parameter (γ = 1.5) was selected to balance pathological signal extraction and physiological variability in sarcopenia COP data. Given the overlap between age-related neuromuscular slowing (i.e., benign sway, linked to age-related neuromuscular degradation [31]) and sarcopenia-driven postural deficits (i.e., pathological misalignments, stemming from muscle function deficits [29]), γ = 1.5 prioritized subtle temporal misalignments (e.g., delayed hip corrections, consistent with sarcopenia-related impaired postural coordination [21]) while minimizing overreaction to large but benign sway. This balance aligns with DTW’s design logic [20] to distinguish pathology-specific patterns, as reflected by its superior performance in capturing meaningful variations.

For LSTM, a 2-unit architecture was chosen to balance performance, efficiency, and interpretability in our small cohort. While deeper models (e.g., 16-unit) showed marginal gains, the 2-unit design reduced overfitting risk (12 vs. 96 features) and enabled 3.6× faster inference—critical for clinical translation—without sacrificing mechanistic interpretability.

#### 5.3.2. Performance Gap and Mechanistic Advantages

The integrated framework captures temporal misalignment (via DTW, quantifying delays in hip corrections), long-range sequential dependencies (via LSTM, encoding cyclic hip stabilizer deficits), and dynamic variability (detecting irregular instability trends). This synergy allows differentiation of reserve states even among patients with similar AWGS 2019 severity scores.

Notably, this advantage aligns with real-world clinical needs:(1)Community screening: As a preliminary triage tool, individuals with DTW-COP-CX in the top 30% of sarcopenia patients (semi-tandem stance) could be flagged for gold standard AWGS 2019 assessment.(2)Post-diagnosis intervention: For confirmed patients, those with (*P*(*Y* = 1) > 0.7 (depleted reserve) might receive intensive hip-focused training, while others receive routine monitoring. Thresholds require multi-center validation but illustrate actionable stratification.

These thresholds require multi-center validation but illustrate actionable stratification. This framework is designed to complement, not replace, gold standards: it fits into existing workflows as a post-diagnosis assessment (after AWGS 2019 confirmation) to refine intervention prioritization, rather than as a standalone diagnostic tool.

Beyond these practical advantages in clinical translation, our framework also stands out for its specificity to sarcopenia-related compensatory reserve—distinguishing it from broader balance assessment methodologies that target general age-related declines. This distinction is particularly evident when compared to existing COP-based approaches, such as that of Alcan [13]; while Alcan [13] targets general age-related decline, our healthy template-anchored DTW explicitly quantifies individual deviations from normative control within sarcopenia patients. This enables not only superior binary classification but, uniquely, simultaneous stratification of compensatory reserve via (*P*(*Y* = 1) variability—a dimension irreplaceable for precision management.

### 5.4. Model Interpretability Considerations

SHAP analysis provided consistent feature importance patterns across the high-performing KNN, RF, and ET models, reinforcing the biological plausibility of key features like DTW distances. Interpretation was more challenging for the SVM (RBF kernel) due to its inherent nonlinearity. For practical translation, models like RF offer a favorable balance of performance and interpretability.

### 5.5. Limitations and Future Directions

Sample size constraints (*N* = 82) and single-center recruitment reflect the challenges of studying AWGS 2019-defined sarcopenia, a population with specific diagnostic criteria that limits broad enrollment. This restricts generalizability—particularly for subtle correlations (e.g., DTW-COP-LX with ASMI), which may underestimate true associations due to limited statistical power and require larger, multi-center cohorts to validate. Future multi-center validation can leverage the wide adoption of clinical force platforms (e.g., Zebris, Kistler) in Chinese secondary hospitals, as these devices are already used for geriatric balance testing—this will enable efficient sample expansion without additional hardware investment. However, as noted in Section 5.1, sarcopenia patients exhibited substantial intra-group variability in ST stance ((*P*(*Y* = 1): CV = 22.7%), and this variability was consistently captured across classifiers, significantly associated with DTW features (reflecting deviations from healthy templates; Section 5.3.1). This provides strong preliminary evidence that the observed variations are not random noise.

As a proof-of-concept, our focus was to validate the feasibility of using temporal features to detect compensation-related differences (rather than large-scale population verification). These findings sufficiently demonstrate the framework’s feasibility to quantify compensatory reserve, laying the groundwork for larger studies to refine correlation strength and clinical thresholds. Notably, our design (feature selection with *N*/*p* ≥ 10, SMOTE, fixed LSTM) mitigated overfitting risks, further supporting the reliability of observed trends.

Furthermore, while the classification performance and observed prediction probability variability provide strong evidence for capturing compensatory reserve, robust clinical validation establishing these metrics as predictors of functional outcomes (e.g., falls, disability progression) or intervention response necessitates prospective longitudinal studies in larger cohorts.

Additional limitations must be addressed to advance clinical translation:(1)Generalizability: Single-center data and a lack of public sarcopenia COP datasets restrict broader applicability. Future multi-center studies should include diverse populations (e.g., varying comorbidities, ethnicities) and integrate gold standard diagnostics to validate findings across demographics;(2)Mechanistic gaps: The COP-based approach captures postural patterns but lacks direct links to neuromuscular mechanisms. Integrating surface EMG (e.g., recording hip stabilizer activity during semi-tandem stance) could correlate DTW/LSTM features with muscle activation timing, validating “hip control deficit” hypotheses and differentiating sarcopenia from other balance disorders.(3)Assessment scope: Static standing (e.g., semi-tandem stance) is a well-established method for quantifying postural control via COP trajectory analysis, with prior validation in neuromuscular research for capturing subtle balance variations linked to muscle weakness. Our focus on static conditions prioritizes measuring slow, sustained dynamics that are central to compensatory reserve—specifically, the delayed or altered sway adjustments (captured via DTW) that reflect an individual’s capacity to mask deficits over time. These slow-adjustment patterns, which are hallmarks of compensatory reserve, may be less detectable in dynamic tasks (e.g., sit-to-stand) where rapid force demands overshadow the gradual neuromuscular adjustments we aim to quantify. While future work could integrate dynamic tasks to explore broader balance mechanisms, such tasks are not central to the current framework, which is designed to isolate static compensatory reserve—their absence does not affect the validity of our findings regarding static-related reserve.(4)Multimodal integration: Standalone COP data cannot disentangle sarcopenia from comorbidities (e.g., frailty). Prioritizing synchronous measurements—such as bioimpedance-derived ASMI (linking to muscle mass loss) and sEMG—will bridge postural performance with sarcopenia’s core pathology.

Larger cohorts will also enable exploring end-to-end training of temporal models (e.g., fine-tuning LSTM/Transformers) to capture deeper patterns—complementing our current lightweight design (aligned with small-sample constraints [21]). Such validation is critical to solidify diagnostic relevance, balancing model depth with interpretability via explainable AI.

Notably, this study focused exclusively on definite sarcopenia, with the framework offering preliminary insights into “compensatory reserve”—an understudied dimension. Temporal patterns (e.g., delayed sway recovery) may differentiate preserved vs. declining compensation, but this requires confirmation in larger cohorts. Given our cross-sectional design, longitudinal studies are needed to determine if these features predict functional decline or intervention response.

## 6. Conclusions

In conclusion, this study presents a robust computational tool that translates COP time-series from a routine clinical force platform into a quantifiable metric of compensatory reserve in compensatory reserve among AWGS 2019-defined sarcopenia patients. The framework’s high discriminative performance (accuracy = 0.84, ROC-AUC = 0.86) in this posture serves as direct evidence for its ability to quantify this reserve gradient, as further substantiated by significant intra-group variability in model outputs.

Key contributions include the following:

Semi-tandem stance as a sensitive condition: By amplifying hip-driven postural demands, it uncovers compensated deficits masked in simple stances, adding nuance to standard evaluations. This aligns with biomechanical theory linking narrow bases of support to hip-dependent control (Section 5.1);

Temporal features’ superiority: Integrating DTW (local misalignments), LSTM (sequential dependencies), and statistical metrics (global trends) outperformed kinematic features by 18% in accuracy, capturing dynamic patterns (e.g., delayed hip corrections) critical for distinguishing reserve variations (Section 5.2);

Clinical translatability: As a complement to gold standards, it enables two scenarios—rapid community screening (flagging individuals with COP deviations from healthy templates) and post-diagnosis stratification (guiding prioritized intervention for severe reserve depletion).

Limitations include a small single-center sample, static stance focus, and unvalidated neuromuscular links—reinforcing its adjunctive role. Future work should validate in multi-center cohorts, integrate dynamic tasks and EMG (to link features to muscle activation), and refine thresholds for clinical use.

In summary, temporal COP analysis in semi-tandem stance shows promise as a complementary tool for exploring compensatory reserve in sarcopenia, with value in enhancing risk stratification by distinguishing intra-group reserve variations among diagnosed individuals, where higher reserve may indicate lower short-term functional decline risk, alongside existing gold standards.

## Figures and Tables

**Figure 1 bioengineering-12-01143-f001:**
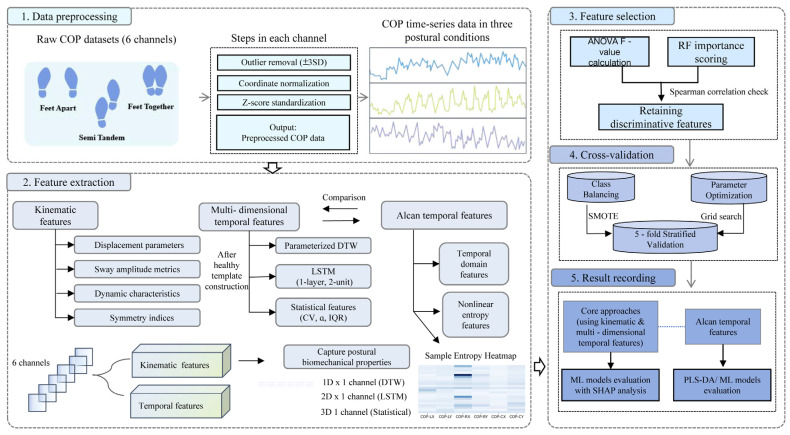
Bioengineering framework for compensatory reserve assessment. The workflow illustrates the pipeline from raw center-of-pressure (COP) signal acquisition to the final model interpretation.

**Figure 2 bioengineering-12-01143-f002:**
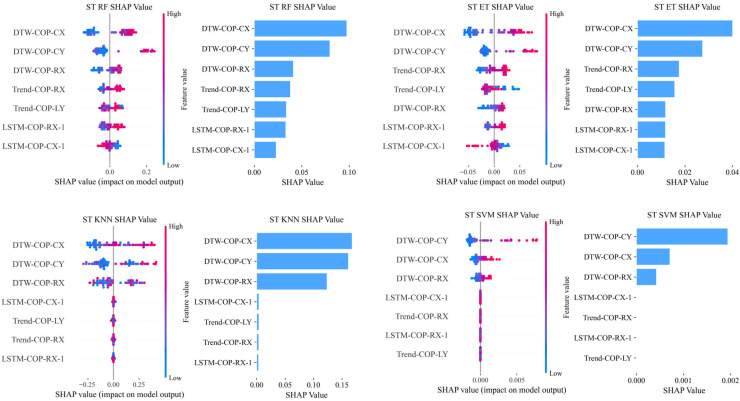
SHAP summary plots of multidimensional temporal features for sarcopenia classification in semi-tandem stance. Results are shown for the Random Forest (RF), Extra Trees (ET), K-Nearest Neighbors (KNN), and Support Vector Machine (SVM) models.

**Figure 3 bioengineering-12-01143-f003:**
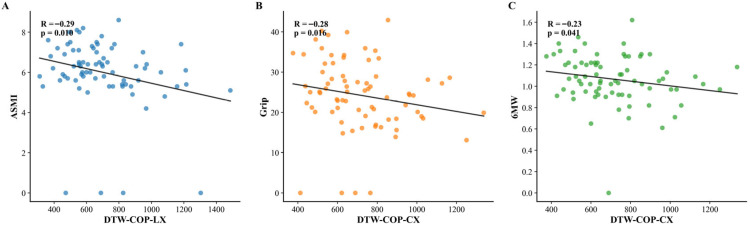
Scatter plots showing significant correlations between key temporal COP features and clinical indicators of sarcopenia in semi-tandem stance. (**A**) DTW-COP-LX vs. appendicular skeletal muscle index (ASMI); (**B**) DTW-COP-CX vs. grip strength; (**C**) DTW-COP-CX vs. 6 min walk (6MW) speed.

**Table 1 bioengineering-12-01143-t001:** Pooled analysis of demographic and clinical characteristics. Comparison between sarcopenia and non-sarcopenia groups across all participants. The bolding indicates statistically significant results.

Characteristic	Sarcopenia	Non-Sarcopenia	*p*-Value
Age (y)	71.04 ± 6.00	68.84 ± 6.34	0.24
Height (cm)	153.80 ± 10.60	155.28 ± 6.91	0.51
Weight (kg)	58.07 ± 8.74	60.46 ± 9.12	0.26
Grip (kg)	20.33 ± 2.50	25.75 ± 3.20	**<0.001**
5TCS time (s)	12.89 ± 1.25	8.46 ± 1.05	**<0.05**
6MW (m/s)	0.85 ± 0.09	1.17 ± 0.11	**<0.001**
ASMI (kg/m^2^)	6.08 ± 0.35	6.49 ± 0.45	**<0.05**

**Table 2 bioengineering-12-01143-t002:** Gender-stratified analysis of demographic and clinical characteristics. Comparison between sarcopenia and non-sarcopenia groups evaluated separately in males and females. The bolding indicates statistically significant results.

Gender	Characteristic	Sarcopenia	Non-Sarcopenia	*p*-Value
Male	Age (y)	71.75 ± 5.72	69.54 ± 4.93	0.24
	Height (cm)	161.77 ± 6.11	159.43 ± 7.15	0.34
	Weight (kg)	61.62 ± 7.74	65.16 ± 9.07	0.26
	Grip (kg)	25.50 ± 1.90	30.20 ± 2.80	**<0.001**
	5TCS (s)	12.50 ± 1.10	7.90 ± 0.90	**<0.05**
	6MW (m/s)	0.82 ± 0.07	1.19 ± 0.10	**<0.01**
	ASMI (kg/m^2^)	6.80 ± 0.15	7.25 ± 0.30	**<0.01**
Female	Age (y)	70.47 ± 6.35	68.29 ± 7.28	0.33
	Height (cm)	147.42 ± 9.02	152.07 ± 4.72	0.08
	Weight (kg)	55.23 ± 8.67	56.83 ± 7.45	0.14
	Grip (kg)	16.20 ± 1.10	22.30 ± 2.10	**<0.001**
	5TCS (s)	13.20 ± 1.30	8.90 ± 1.00	**<0.05**
	6MW (m/s)	0.88 ± 0.06	1.15 ± 0.11	**<0.001**
	ASMI (kg/m^2^)	5.50 ± 0.18	5.90 ± 0.25	**<0.05**

Note: 5TCS, 5-time chair stand test; 6MW, 6 min walk test; ASMI, appendicular skeletal muscle index. Data presented as mean ± standard deviation.

**Table 3 bioengineering-12-01143-t003:** Posture-specific classification performance (mean ± SD across seven classifiers) for sarcopenia identification using three feature extraction methods.

Feature Method	Postural Conditions	Accuracy	Precision	Recall	F1-Score	ROC-AUC	AUPR
Kinematic	FT	0.63 ± 0.08	0.61 ± 0.11	0.68 ± 0.10	0.63 ± 0.10	0.67 ± 0.13	0.67 ± 0.08
	FA	0.65 ± 0.08	0.65 ± 0.09	0.70 ± 0.08	0.67 ± 0.09	0.71 ± 0.13	0.71 ± 0.09
	ST	**0.66 ± 0.08**	**0.63 ± 0.07**	**0.76 ± 0.10**	**0.68 ± 0.11**	**0.73 ± 0.17**	**0.76 ± 0.09**
Multidimensional Temporal	FT	0.70 ± 0.06	0.69 ± 0.09	0.80 ± 0.14	0.72 ± 0.06	0.66 ± 0.10	0.67 ± 0.11
	FA	0.77 ± 0.07	0.77 ± 0.08	0.81 ± 0.13	0.77 ± 0.08	0.75 ± 0.10	0.74 ± 0.12
	ST	**0.84 ± 0.04**	**0.83 ± 0.07**	**0.89 ± 0.09**	**0.85 ± 0.04**	**0.86 ± 0.05**	**0.85 ± 0.07**
Alcan ’s method (Scenario B)	FT	0.67 ± 0.06	0.74 ± 0.05	0.63 ± 0.10	0.61 ± 0.10	0.63 ± 0.09	0.68 ± 0.07
	FA	0.68 ± 0.05	0.70 ± 0.04	0.74 ± 0.07	0.68 ± 0.06	0.63 ± 0.07	0.63 ± 0.06
	ST	**0.75 ± 0.05**	**0.79 ± 0.06**	**0.75 ± 0.09**	**0.73 ± 0.06**	**0.74 ± 0.05**	**0.78 ± 0.05**

Note: FT, feet-together; FA, feet-apart; ST, semi-tandem; ROC-AUC, area under the receiver operating characteristic curve; AUPR, area under the precision–recall curve. Bold values indicate the best classification performance under the semi-tandem (ST) postural condition across the respective feature extraction methods.

**Table 4 bioengineering-12-01143-t004:** Comparison of classification performance for sarcopenia identification in semi-tandem stance using three feature extraction methods. The bold values highlight the best classification performance of the multidimensional temporal framework in the semi-tandem stance.

Feature Method	Accuracy	Precision	Recall	F1-Score	ROC-AUC	AUPR
Kinematic features	0.66 ± 0.08	0.63 ± 0.07	0.76 ± 0.10	0.68 ± 0.11	0.73 ± 0.17	0.76 ± 0.09
Multidimensional temporal features	**0.84 ± 0.04**	**0.83 ± 0.07**	**0.89 ± 0.09**	**0.85 ± 0.04**	**0.86 ± 0.05**	**0.85 ± 0.07**
Alcan’s method (Scenario B)	0.75 ± 0.05	0.79 ± 0.06	0.75 ± 0.09	0.73 ± 0.06	0.74 ± 0.05	0.78 ± 0.05

**Table 5 bioengineering-12-01143-t005:** Detailed classification performance of individual machine learning models using multidimensional temporal features in semi-tandem stance.

Model	Accuracy	Precision	Recall	F1-Score	ROC-AUC	AUPR
KNN	0.88 ± 0.05	0.88 ± 0.07	0.89 ± 0.04	0.88 ± 0.04	0.90 ± 0.05	0.89 ± 0.06
RF	0.88 ± 0.05	0.88 ± 0.10	0.91 ± 0.08	0.89 ± 0.04	0.91 ± 0.05	0.91 ± 0.08
ET	0.85 ± 0.02	0.82 ± 0.04	0.89 ± 0.11	0.85 ± 0.03	0.89 ± 0.05	0.87 ± 0.09
SVM	0.85 ± 0.04	0.84 ± 0.10	0.89 ± 0.11	0.85 ± 0.03	0.87 ± 0.03	0.89 ± 0.02
LR	0.83 ± 0.04	0.84 ± 0.10	0.84 ± 0.11	0.83 ± 0.04	0.83 ± 0.06	0.80 ± 0.12
NB	0.82 ± 0.03	0.83 ± 0.05	0.82 ± 0.11	0.82 ± 0.04	0.82 ± 0.06	0.82 ± 0.08
DT	0.79 ± 0.07	0.72 ± 0.06	0.96 ± 0.04	0.82 ± 0.05	0.80 ± 0.06	0.80 ± 0.05

**Table 6 bioengineering-12-01143-t006:** Distribution of top predictive features identified by SHAP analysis. Frequency of each multidimensional temporal feature type appearing among the top five most important features across high-performing models (KNN, RF, ET, SVM) in three postural conditions.

Feature Type	Frequency in top 5 Important Features		
	FT	FA	ST
DTW features	0	0	9
LSTM features	4	5	1
Statistical features	8 (4 trend slope features)	9 (8 trend slope features)	5 (5 trend slope features)

**Table 7 bioengineering-12-01143-t007:** Intra-group variability in compensatory reserve among sarcopenia patients. Distribution of the model-derived prediction probability *P*(*Y* = 1) across different postural conditions, characterized by range, mean ± standard deviation (SD), and coefficient of variation (CV).

Postural Conditions	Range	Mean ± SD	CV
FT	(0.34, 0.84)	0.48 ± 0.10	20.3%
FA	(0.37, 0.79)	0.55 ± 0.14	25.3%
ST	(0.31, 0.78)	0.58 ± 0.13	22.7%

Note: *P*(*Y* = 1), predicted probability of being classified as sarcopenia; CV, coefficient of variation.

## Data Availability

The data presented in this study are available on request from the corresponding author. The data are not publicly available due to privacy and ethical restrictions.

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
