# Peer review of "Capturing Compensatory Reserve in Sarcopenia: A Bioengineering Framework for Multidimensional Temporal Analysis of Center-of-Pressure Signals"

_bioengineering, 2025, doi:10.3390/bioengineering12111143_

Round 1
Reviewer 1 Report
Comments and Suggestions for Authors
The review of the submission bioengineering-3926488, entitled “Capturing Compensatory Reserve in Sarcopenia: A Bioengineering Framework for Multidimensional Temporal Analysis of Center-of-Pressure Signals”:
- The main question addressed by the research – the development of the methods to treat sarcopenia.
- The paper addresses the necessity of the improvement of data collection to better describe and monitor treatment
- The quality of English does not require any improvements
- The conclusions are consistent with the evidences and arguments presented.
- All main questions posed in the Introduction were addressed by collected and demonstrated data.
- Statistical significance of the conclusions looks accepted.
- In comparison with other published material the submission adds well checked ideas of age-caused sarcopenia description.
- The references are appropriate
- Additional comments on figures – on ALL of them text on the images looks INVISIBLE and require enlargement for the sake of readers convenience
Conclusion: Minor revision
Author Response
Response to Reviewer 1:
We sincerely thank the reviewer for their positive and encouraging feedback on our work.
Comment 1: "Additional comments on figures – on ALL of them text on the images looks INVISIBLE and require enlargement for the sake of readers convenience."
Response 1: We sincerely apologize for this oversight and deeply appreciate the reviewer for bringing it to our attention. We have regenerated all figures (Figure 1, 2, and 3) with larger font sizes, higher resolution, and improved contrast to ensure all text, labels, and data points are clearly visible. The revised figures are now submitted with the manuscript.

Reviewer 2 Report
Comments and Suggestions for Authors
The manuscript “Capturing Compensatory Reserve in Sarcopenia: A Bioengineering Framework for Multidimensional Temporal Analysis of Center-of-Pressure Signals” proposes a novel bioengineering framework for the analysis of Center-of-Pressure (COP) signals to quantify "compensatory reserve" in older adults with sarcopenia. The core hypothesis is that conventional static or simple time-domain COP measures fail to detect subtle postural deficits because they are masked by effective compensatory strategies. The study utilizes a multidimensional temporal analysis technique to extract more nuanced features from the COP data, aiming to provide a more sensitive diagnostic tool for sarcopenia-related balance impairment.
The study’s major strengths include its novelty and clinical relevance, as quantifying “compensatory reserve” addresses a key gap in detecting subtle deficits missed by conventional tests. Its use of a sophisticated “multidimensional temporal analysis” framework demonstrates strong bioengineering insight beyond standard COP metrics. Additionally, the paper is well-structured, with clear objectives linking the identified problem to an innovative analytical solution.
Key Areas for Improvement:
- Introduction - The text effectively establishes the clinical motivation, scientific gap, and methodological innovation of the study.
- Data Collection and Verification – line 117 – “Ethics approval (Helsinki Declaration) and informed consent were obtained.” Please add the name of the institution and the document number.
Line 122 – Please provide more detailed specifications for the platform – dimensions – sensors.
Lines 130–137 - “Tests were administered by nurses who received 1-hour training…”
Suggestion: “All assessments were conducted by trained nurses who received standardized 1-hour instruction in posture setup (e.g., semi-tandem stance) and COP recording procedures.” Then, please clarify trial structure concisely: “Each participant completed three 20-s trials (10-s rests) in three postural conditions—feet-together (FT), feet-apart (FA), and semi-tandem (ST). Unstable trials were repeated once; persistent instability led to exclusion.”
Line 132-133 – Please describe each position (FT, FA, ST) and how exactly the feet were positioned.
- The methods section is clear, detailed, and well-structured. It effectively presents the analytical framework, justifies the feature extraction approach, and links compensatory reserve to model interpretation. The five-step pipeline, from preprocessing to interpretation, is comprehensive and aligns with standard practices in computational biomechanics research. I have no comments for this section.
- Results. This section is clearly written and well-organized. My comments pertain primarily to the figure and table captions. These should be more precise and include explanations for all abbreviations used.
At line 437, please complete the figure reference with the appropriate figure number.
Additionally, Figure 2 is currently illegible—the legend is missing, the font size is too small, and the overall image quality is poor. Figure 3 is clearer but would also benefit from a larger font size to improve readability.
Literature - no comments.
Final Recommendation: minor revision.
Author Response
Response to Reviewer 2:
Dear Reviewer,
We are profoundly grateful to the reviewer for their thorough and thoughtful review, and for recognizing the novelty and clinical relevance of our work.
Comment 1 (Data Collection - Ethics): "Data Collection and Verification – line 117 – “Ethics approval (Helsinki Declaration) and informed consent were obtained.” Please add the name of the institution and the document number."
Response 1: We thank the reviewer for highlighting this need for clarity. We have revised the sentence to include the full ethical oversight information, including the continuing review approval for the supplementary work conducted in 2024 (Lines 122-126):
"The study was conducted in accordance with the Declaration of Helsinki and approved by the Medical Ethics Committee of West China Hospital of Sichuan University (Initial Approval No. 2021[96]; Continuing Review Approval No. 2024[1305]). Informed consent was obtained from all participants."
Comment 2 (Data Collection - Platform Specs): "Line 122 – Please provide more detailed specifications for the platform – dimensions – sensors."
Response 2: We are grateful for this suggestion to enhance methodological transparency. We have added the requested technical specifications for the Zebris FDM platform. Furthermore, to enhance methodological transparency, we have also integrated a concise description of the sarcopenia diagnostic criteria and tools used in the cohort verification, as suggested by the reviewer's overall focus on clarity (Lines 132-136):
"COP data were collected during quiet standing using a zebris FDM-system (zebris Medical GmbH, Germany). Measurements were performed on an FDM 1.5 platform (sensor area: 149.0 × 54.2 cm) housing 11,264 capacitive sensors. The system recorded raw COP trajectories at 100 Hz with a spatial resolution of 1 mm through its native WinFDM software, providing high-fidelity data for extracting both kinematic and proposed multidimensional temporal features."
Comment 3 (Data Collection - Procedure): "Lines 130–137 - “Tests were administered by nurses who received 1-hour training…”
Suggestion: “All assessments were conducted by trained nurses who received standardized 1-hour instruction in posture setup (e.g., semi-tandem stance) and COP recording procedures.” Then, please clarify trial structure concisely: “Each participant completed three 20-s trials (10-s rests) in three postural conditions—feet-together (FT), feet-apart (FA), and semi-tandem (ST). Unstable trials were repeated once; persistent instability led to exclusion."
Response 3: We thank the reviewer for this excellent suggestion to improve conciseness and clarity. We have revised the paragraph accordingly (Lines 146-156):
"All assessments were conducted by trained nurses who received standardized 1-hour instruction in posture setup and COP recording procedures. Each participant completed three 20-s trials (with 10-s rests between trials) under three postural conditions: Feet-Together (FT), Feet-Apart (FA), and Semi-Tandem (ST). Unstable trials were repeated once; persistent instability led to exclusion."
Comment 4 (Data Collection - Posture Description): "Line 132-133 – Please describe each position (FT, FA, ST) and how exactly the feet were positioned."
Response 4: We appreciate the reviewer's request for precision. We have added a clear and precise description of the foot positions for each posture as per our standardized protocol (Lines 149-154):
"The foot placement for each condition was standardized as follows:
Feet-Together (FT): Participants stood with the medial borders of both feet touching.
Feet-Apart (FA): Participants stood with their feet separated at an interior distance of 10 cm.
Semi-Tandem (ST): Participants stood with the heel of one foot placed beside the arch of the contralateral foot, creating a narrowed base of support."
Comment 5 (Results - Figures): "Results. This section is clearly written and well-organized. My comments pertain primarily to the figure and table captions. These should be more precise and include explanations for all abbreviations used. At line 437, please complete the figure reference with the appropriate figure number. Additionally, Figure 2 is currently illegible—the legend is missing, the font size is too small, and the overall image quality is poor. Figure 3 is clearer but would also benefit from a larger font size to improve readability."
Response 5: We extend our sincere thanks to the reviewer for these crucial observations and apologize for the shortcomings in the original figures. We have addressed each point in detail:
Figure and Table Captions: We have thoroughly revised all figure and table captions to ensure they are more precise and descriptive. All non-standard abbreviations used within the figures and tables are now clearly defined in their respective captions.
Figure Reference: Thank you for catching this omission. The incomplete sentence has been corrected to reference Figure 2.
Figure 2 Legibility: We sincerely apologize for the poor quality of the original Figure 2. We have completely regenerated this figure with high resolution and enhanced overall clarity to ensure all elements, including text and data points, are easily interpretable.
Figure 3 Readability: As per the reviewer's suggestion, we have also regenerated Figure 3 with larger font sizes for all axes labels, ticks, and annotations to enhance its readability.
Once again, we deeply appreciate your insightful and detailed feedback, which has significantly strengthened our manuscript.
